# Cranial anatomy of *Emeroleter levis* and the phylogeny of Nycteroleteridae

**Kayla D. Bazzana-Adams**[ID][1]*, **Mark J. MacDougall**[2], **Jörg Fröbisch**[ID][2]

**1** Faculty of Veterinary Medicine, University of Calgary, Calgary, Alberta, Canada, **2** Museum für Naturkunde, Leibniz-Institut für Evolutions-und Biodiversitätsforschung, Berlin, Germany

* kayla.bazzanaadams@ucalgary.ca

**Data Availability Statement:** Raw CT data are available on MorphoSource (https://www.morphosource.org/projects/000618423).

**Funding:** This work was supported by funding from the German Research Foundation (DFG

## Abstract

Among the diverse basal reptile clade Parareptilia, the nycteroleters are among the most poorly understood. The interrelationships of nycteroleters are contentious, being recovered as both monophyletic and paraphyletic in different analyses, yet their anatomy has received little attention. We utilized x-ray computed tomography to investigate the skull of the nycteroleterid *Emeroleter levis*, revealing aspects of both the external and internal cranial anatomy that were previously unknown or undescribed, especially relating to the palate, braincase, and mandible. Our results reveal a greater diversity in nycteroleter cranial anatomy than was previously recognized, including variation in the contribution of the palatal elements to the orbitonasal ridge among nycteroleters. Of particular note are the unique dentition patterns in *Emeroleter*, including the presence of dentition on the ectopterygoid, an element which is typically edentulous in most parareptiles. We then incorporate the novel information gained from the computed tomography analysis into an updated phylogenetic analysis of parareptiles, producing a fully resolved Nycteroleteridae and further supporting previous suggestions that the genus *'Bashkyroleter'* is paraphyletic.

## Introduction

Parareptilia is a taxonomically and morphologically diverse clade of basal sauropsids with an exceptional fossil record across the end-Permian mass extinction [1–3]. The clade is generally considered to be the sister group to Eureptilia [4,5], the clade that includes all living reptiles, though other recent work has recovered the alternative position of Parareptilia nested within Diapsida [6] and some analyses recover Parareptilia as paraphyletic [7]. Within parareptiles, the Russian nycteroleters are particularly poorly understood. First named by Efremov [8], nycteroleters are frequently recovered as sister to pareiasaurs within Parareptilia [9–13]. The interrelationships of nycteroleters themselves, however, is contentious, being recovered as monophyletic in some analyses and paraphyletic in others [13,14]. Ivakhnenko [14] placed the Russian nycteroleters in a paraphyletic grouping with lanthanosuchids, whereas Tsuji et al. [13] found a monophyletic grouping using parsimony but a paraphyletic one using Bayesian inference. Even in those analyses which recover nycteroleters as monophyletic, the interrelationships within the group remain largely unresolved [3,5,13].

project No. 2457/9-1) and a scholarship to KBA from the Humboldt Internship Program. The funders had no role in study design, data collection and analysis, decision to publish, or preparation of the manuscript.

**Competing interests:** The authors have declared that no competing interests exist.

Despite the ongoing uncertainty regarding the phylogenetic positioning of the nycteroleters, the anatomy of Russian nycteroleters has received little attention, with *Macroleter poezicus*, *Emeroleter levis*, and an unnamed taxon from South Africa being the only taxa whose anatomy have been recently re-described [9,13,15]. *Emeroleter* is the youngest known nycteroleter and the only taxon known from the late Permian [13]. A re-description of *Emeroleter* by Tsuji et al. [13] greatly expanded the current body of knowledge regarding nycteroleter anatomy, particularly that of the postcranium; however, very little has been known of the anatomy of the palate, braincase, or mandible of *Emeroleter*. Given the problematic nature of nycteroleter phylogeny, and the growing emphasis on the utility of braincase characters in interpretations of Palaeozoic tetrapod relationships [16], clarifying the anatomy of these poorly understood regions can greatly inform discussions of nycteroleter evolution and relationships.

Here we employ x-ray computed tomography (CT) to examine the skull of the nycteroleterid *Emeroleter levis*, revealing new features of both the external and internal cranial anatomy that were previously unknown or undescribed, especially as regards the palate, braincase, and mandible. The new information gained from this allowed us to conduct an updated phylogenetic analysis of parareptiles, to better investigate the historically poorly resolved clade Nycteroleteridae.

## Materials and methods

No permits were required for the described study, which complied with all relevant regulations.

### Scanning and segmentation

KPM uncat/E2 was scanned using the x-ray computed tomography equipment (Phoenix | x-ray Nanotom | s) at the Museum für Naturkunde in Berlin. Scan parameters were set at 160 kV voltage and 100 μA current. A total of 1857 projections were taken over 2h7m37s, with a final voxel size of 0.0257 mm x 0.0257 mm x 0.0257 mm. Manual segmentation of all elements was performed in Amira 3D 2021.2.

### Phylogenetic analysis

New character data was coded into the character taxon matrix of Cisneros et al. [5], which was derived from the matrix of MacDougall et al. [4]. The phylogenetic analysis was performed in PAUP 4.0a169 [17] with parsimony being designated as the optimality criterion for the phylogenetic analysis and minimum branch lengths of less than zero set to collapse. *Seymouria* was designated as the outgroup taxon. A heuristic search with tree bisection and reconnection (TBR) branch swapping was conducted, the addition sequence algorithm was set to random (10000 replicates), holding 10 trees at each step. A bootstrap analysis, using fast stepwise addition (10000 replicates), was conducted to determine the support values for the recovered nodes. The data matrix and associated character list can be found in the supplementary information.

Updated character codings for *Emeroleter levis* are as follows: paired postparietals (8)? →0, jugal anterior process extends to at least the level of the anterior orbital rim in this specimen (18) 0→0/1, jugal suborbital ramus is dorsoventrally broad (19) 1→0, absence of heterodont dentition on dentary (36)? →1, quadratojugal anterior extent reaches the posterior margin of the orbit in this specimen (40) 1→0/1, choana orientated parallel to maxilla (61)? →0, cranioquadrate space is large (64)? →1, pterygoid anterior extent reaches the posterior end of the choana (65)? →0, absence of dentition on the quadrate flange of the pterygoid (70)? →0, deep excavation on posterolateral surface of the quadrate ramus (71) 0→1, quadrate ramus of

pterygoid is continuous with transverse flange (72)? →1, presence of lateral pocket between quadrate flange and transverse flange of pterygoid (73)? →1, ectopterygoid is present (74)? →0, ectopterygoid teeth are present (75)? →0, ectopterygoid does not contribute to lateral portion of transverse flange of pterygoid (76)? →0, supraoccipital is plate like with sagittal crest (87)? →2, anteroposteriorly expanded paroccipital process (88)? →1, paroccipital process directed primarily laterally (89)? →0, paroccipital process to skull roof contact is present (90)? →1, parabasisphenoid to basioccipital contact is present (95)? →0, parabasisphenoid and basioccipital contribute extensively to the ventral surface of the braincase (97)? →0, meckelian fossa faces dorsally (107)? →1, short meckelian fossa (108)? →1, single coronoid element (111)? →1, prearticular does not extend beyond coronoid eminence (112)? →1, retroarticular process composed of only articular (115)? →0, lateral shelf present on articular (116)? →1, coronoid process is high (117)? →1, coronoid process is composed solely of coronoid (118)? →0, snout shape wider than tall (173) 1→0.

## Institutional abbreviations

**KPM**, Vyatka Paleontological Museum, Kirov, Russia.

## Systematic palaeontology

AMNIOTA Haeckel, 1866 [18]
 REPTILIA Laurenti, 1768 [19]
 PARAREPTILIA Olson, 1947 [20]
 NYCTEROLETERIDAE Romer, 1956 [21]
 *EMEROLETER LEVIS* Ivakhnenko, 1997 [22]

## Revised diagnosis

Expanded from Tsuji et al. (2012). Medium-sized nycteroleter that possesses the following autapomorphies: maxilla and quadratojugal do not contact one another; maxilla extends to and contributes to the posterodorsal margin of the naris; posteriorly elongated supratemporals that form long, narrow horns; enlarged, unsculptured otic notch almost extending to posterior margin of orbit; posterior end of quadratojugal curves upwards forming a small horn; middle pterygoid denticle ridge stretches from area of basipterygoid joint to posterior edge of choana and does not adjoin vomeropalatine ridge.

## Referred specimen

KPM uncat/E2, mostly complete skeleton and skull of a probable juvenile, preserving majority of skull roof, palate, mandible, and portions of the braincase. The skull of KMP uncat/E2 has undergone dorsoventral crushing.

## Results

### Anatomical description

The skull roof has been described in detail by Tsuji et al. [13]; therefore, we focus on the more poorly known features, primarily the palate, braincase, and mandible.

### Skull roof

The lacrimal is completely excluded from the external naris by the dorsal process of the maxilla (Fig 1). The ventral edge of the orbital rim portion of the lacrimal contacts an ascending

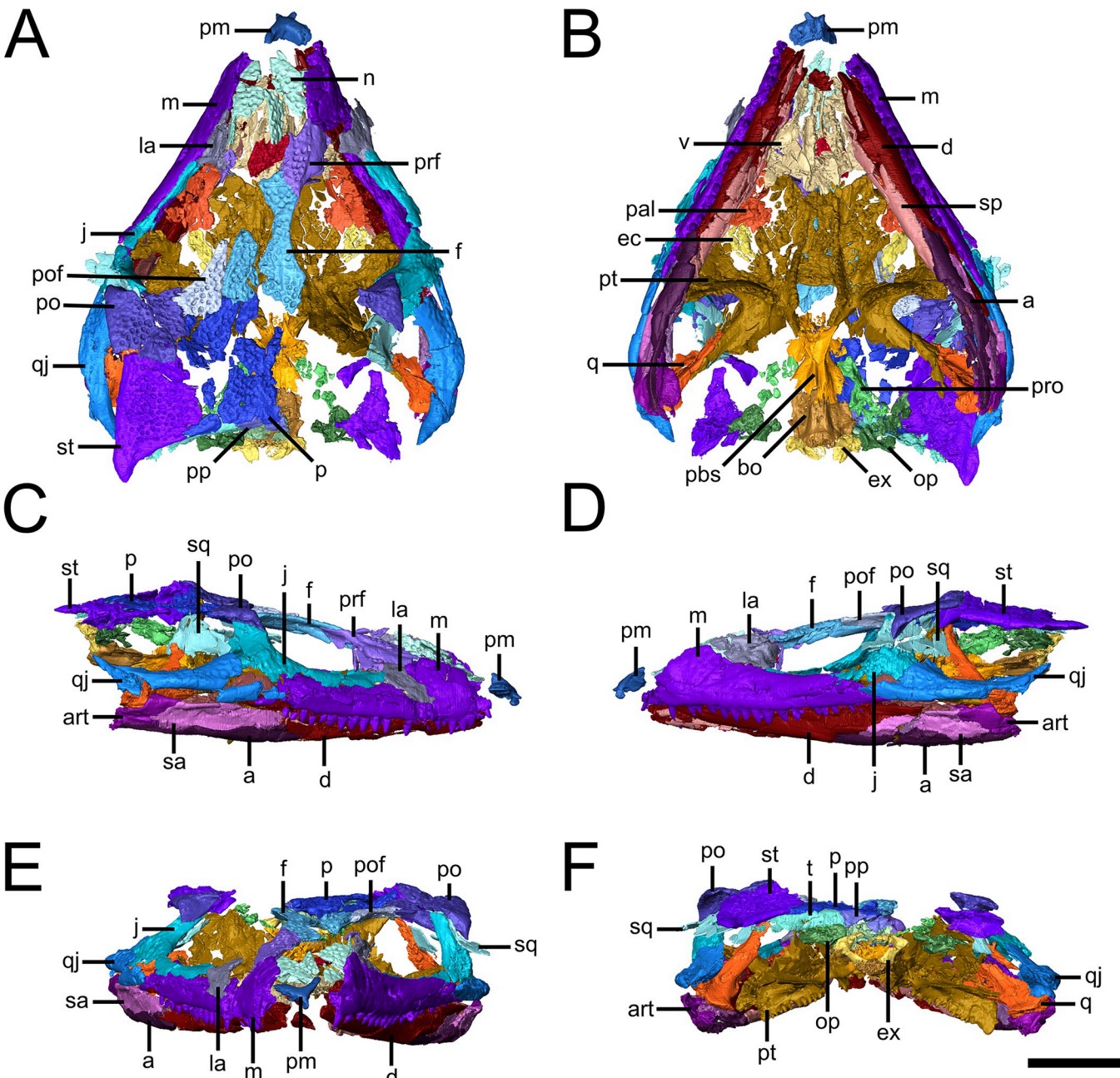

**Fig 1. Skull of the holotype of *Emeroleter levis*.** Virtual rendering of KPM uncat/E2 in **A,** dorsal, **B,** ventral, **C,** right lateral, **D,** left lateral, **E,** anterior, and **F,** posterior views. Abbreviations: **a**, angular; **art**, articular; **bo**, basioccipital; **d**, dentary; **ec**, ectopterygoid; **ex**, exoccipital; **f**, frontal; **j**, jugal; **la**, lacrimal; **m**, maxilla; **n**, nasal; **op**, opisthotic; **pal**, palatine; **pbs**, parabasisphenoid; **pm**, premaxilla; **po**, postorbital; **pof**, postfrontal; **p**, parietal; **pp**, postparietal; **prf**, prefrontal; **pro**, prootic; **pt**, pterygoid; **q**, quadrate; **qj**, quadratojugal; **sa**, surangular; **sp**, splenial; **sq**, squamosal; **st**, supratemporal; **t**, tabular; **v**, vomer. Scale bar equals 1 cm.

process of the palatine. Also of note is that the broad suborbital ramus of the jugal extends forward to reach the anterior orbital rim.

The postparietal is a triangular median element that is largely occipital, making no contribution to the skull roof (Fig 1A). The dorsal portion of the postparietal underlies the parietal, while it forms a slightly ventromedially angled contact with the tabular laterally. Both the

postparietal and tabular are slightly posteroventrally tilted. The dorsal portion of the tabular underlies the lateral half of the parietal and the majority of the supratemporal.

**Palate.** The vomer is roughly triangular, being relatively narrow at its anterior end and broadening posteriorly (Fig 2). The element shares a broad posterior contact with the pterygoid and only a short posterolateral suture with the palatine; the entire posterior edge of the vomer appears to taper posteromedially. A row of denticles runs along the medial edge of the element as a continuation of the denticles running along the medial edge of the palatal ramus of the pterygoid (Fig 2B). Small denticles also appear to run along the ventral surface of the thickened lateral edge of the vomer. The curved flanges visible on the dorsal surface of the vomer are the orbitonasal ridge, reflecting the attachment point for the orbitonasal membrane and forming a concave depression for the paraseptal cartilage of the nasal capsule (Fig 2A) [23]. The posterior half of the lateral margin of the vomer curves medially, forming the antero-medial and medial margin of the choana.

The palatine is also triangular but is relatively much smaller than the vomer (Fig 2). The medial edge curves slightly medially into an indentation in the lateral edge of the palatal ramus of the pterygoid. The thickened anterior end rises dorsally to contact the lacrimal, thus partially flooring the orbit. A thickened ridge bearing denticles runs anterolaterally across the palatine as a continuation of a similarly oriented denticulate ridge on the pterygoid.

The ectopterygoid as preserved in KPM uncat/E2 is a small, roughly oval element situated between the palatine and the quadrate and palatal rami of the pterygoid (Fig 2). An

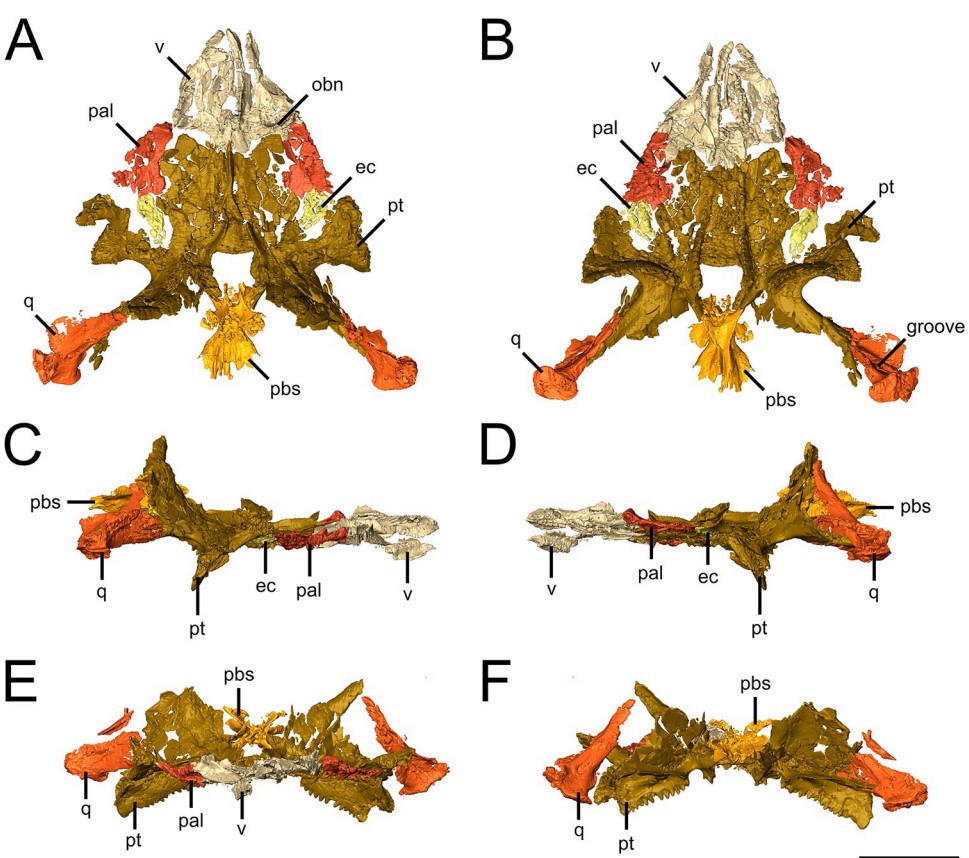

**Fig 2. Palate of the holotype of *Emeroleter levis*.** Isolated views of the palate of KPM uncat/E2 in **A,** dorsal, **B,** ventral, **C,** right lateral, **D,** left lateral, **E,** anterior, and **F,** posterior. Abbreviations: **ec**, ectopterygoid; **obn**, orbitonasal ridge; **pal**, palatine; **pbs**, parabasisphenoid; **pt**, pterygoid; **q**, quadrate; **v**, vomer. Scale bar equals 1 cm.

anterolaterally directed row of denticles is visible on its ventral surface, a feature that is rare among parareptiles (Fig 2B) [24].

The pterygoid is a complex, triradiate element that forms the majority of the palate (Fig 2). The anteriorly directed palatal ramus is broad for most of its length, tapering quickly to a blunt anterolaterally directed point between the vomer and palatine. The interpterygoid vacuity is anteroposteriorly short and mediolaterally broad, resulting in a roughly square outline in ventral view; the anterior and anterolateral margins of the vacuity are formed by the palatal rami of the pterygoid, while the posterior margin is formed by the parabasisphenoid. The ventral surface of the palatal ramus bears a broad field of very small denticles and a prominent ridge running anteriorly from the anteroventral midpoint of the interpterygoid vacuity (Fig 2B); this ridge continues anteriorly onto the vomer. A prominent sharp ridge extends anteromedially along the dorsal surface of the palatal ramus, converging medially to form a single ridge that continues anteriorly for the full length of the element (Fig 2A). It is possible that this ridge continues onto the vomer, but the latter element is too fragmented to confidently identify any such dorsal ridge.

The laterally extending transverse flange has pronounced sharp denticles along its ventral ridge (Fig 2B). The lateral edge of the transverse flange extends ventrally below the level of the maxilla and dorsally to contact the maxilla and jugal (Fig 2C and 2D). The ventral ridge continues posterolaterally into a thin edentulous flange that forms the ventral portion of the quadrate ramus, which in turn has an extensive suture with the pterygoid flange of the quadrate. A large lateral excavation is found below the ridge formed by the transverse flange and quadrate ramus (Fig 2B). An ascending flange extends from the basicranial articulation along the quadrate ramus.

The right quadrate is preserved in contact with the pterygoid with a largely complete dorsal lamella and associated pterygoid ramus, whereas the left quadrate is disarticulated from the pterygoid and lacks much of the dorsal lamella (Fig 2). The preserved portion of the dorsal lamella is roughly trapezoidal in shape and extends anterodorsally to contact the squamosal, and a pronounced groove is visible along the ventral portion of the posterior surface of the dorsal process. The pterygoid ramus of the element continues anteriorly along the same axis as the quadrate ramus of the pterygoid, such that the articular surface of the quadrate is oriented posteroventrolaterally. The flat oval shaped articular surface is well developed and is mediolaterally broader than it is anteroposteriorly long. The pterygoid ramus broadly overlaps the anterior surface of the quadrate ramus of the pterygoid for just over half its length.

**Braincase.** The parasphenoid and basisphenoid are fully fused into an hourglass-shaped parabasisphenoid in which the posterior portion is slightly anteroposteriorly longer than the anterior portion (Fig 3). There is no indication of a cultriform process, as is typical for nycteroleterids [9,22]. The basipterygoid processes project anterolaterally and are firmly sutured to their respective pterygoids. Immediately posterior to the basicranial articulation, mediolaterally compressed clinoid processes extend posterodorsally towards the prootics where they end in unfinished bone. The dorsum sellae is not preserved, likely reflecting the presumed immaturity of the specimen [13]. The abducens nerve (CN.VI) likely passed through the dorsomedial edge of the clinoid processes but incomplete ossification prevents identification of which, if any, of the possible foramina could reflect the passage of that nerve. The ventral surface of the parabasisphenoid bears a trough extending posteriorly from the "waist" of the element, though the narrowness of this trough is exaggerated by the breakage of the element along its midline (Fig 3B). The posterior internal carotid foramen is visible on the lateral surface of the element, at its approximate anteroposterior midpoint.

The basioccipital is broad, flat, and anteroposteriorly elongated, forming a broad overlapping contact with the parabasisphenoid anteriorly (Fig 3). The dorsal surface is slightly

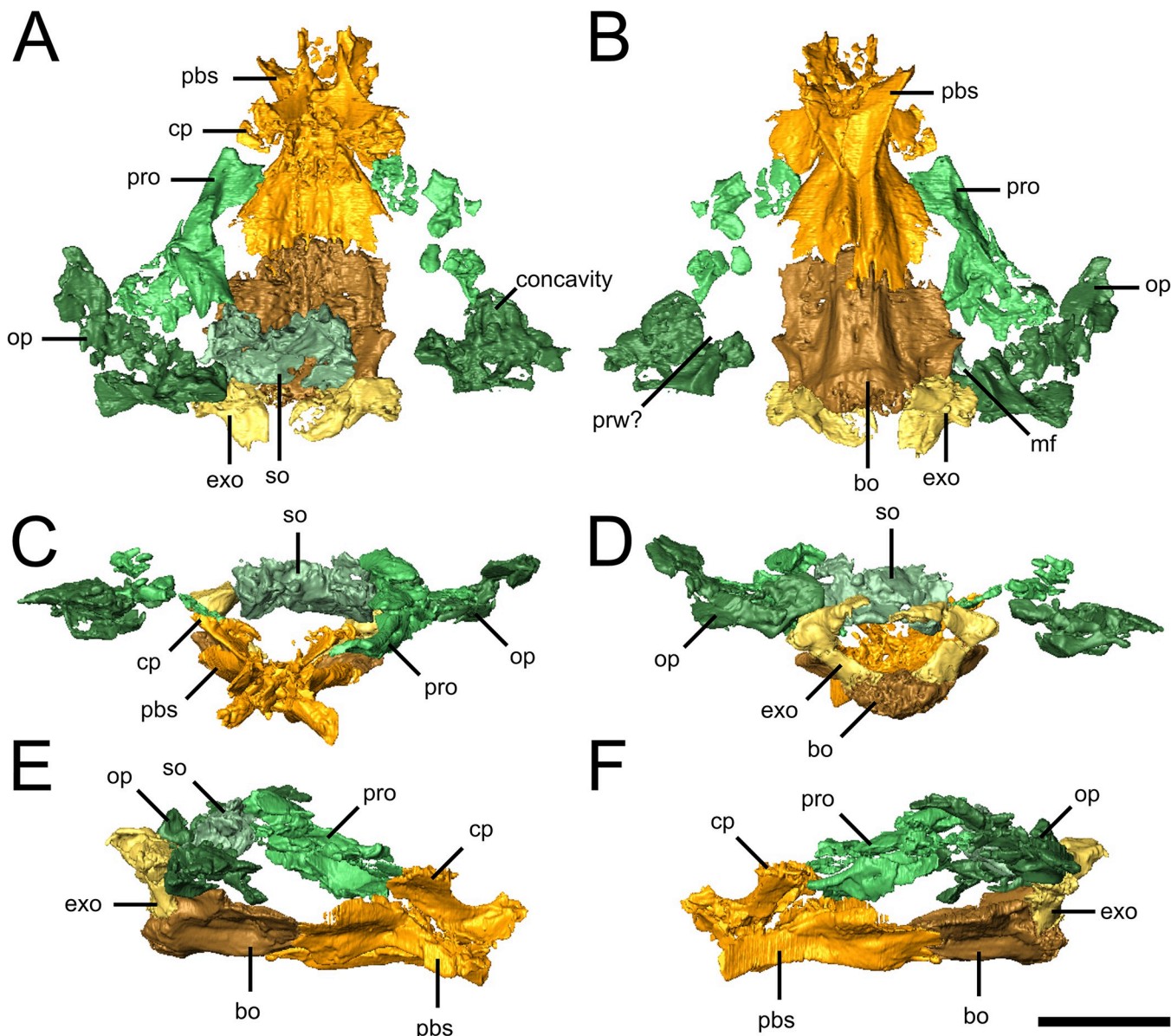

**Fig 3. Braincase of the holotype of *Emeroleter levis*.** Isolated views of the braincase of KPM uncat/E2 in **A,** dorsal, **B,** ventral, **C,** anterior, **D,** posterior, **E,** right lateral, and **F,** left lateral views. Abbreviations: **bo**, basioccipital; **cp**, clinoid process; **exo**, exoccipital; **mf**, metotic foramen; **op**, opisthotic; **pbs**, parabasisphenoid; **prw**, pressure relief window; **pro**, prootic; **so**, supraoccipital. Scale bar equals 5 mm.

concave, with the lateral edges flattened and dorsolaterally directed, likely forming the ventral margin of the fenestra vestibuli. The basioccipital floors the foramen magnum and forms the entirety of the occipital condyle, although the exact nature of the condyle cannot be determined due to the poor ossification of the posterior end of the element. Immediately anterior to its contact with the exoccipitals, the lateral margins of the basioccipital exhibit rounded embayments, which likely reflect the metotic foramen, the morphology of which appears similar to that seen in *Macroleter* [10].

The L-shaped exoccipitals form the lateral walls and roof of the foramen magnum (Fig 3D); unlike in *Macroleter* [9], they do not meet ventrally to exclude the basioccipital from the foramen magnum. The occiput is shifted posteriorly relative to the skull roof due to crushing of

the specimen, but given the extent of the medial projections of the dorsal processes of the exoccipitals, it is likely that they would have excluded the supraoccipital from the foramen magnum.

The prootic is a simple, posteroventrally concave element that frames the anterior portion of the otic capsule, housing at least the anterior portion of the anterior semicircular canal and possibly that of the lateral canal as well (Fig 3). The element appears to have contacted the clinoid processes of the parabasisphenoid anteriorly and the supraoccipital posterodorsally; the nature of the contact with the opisthotic cannot be determined due to the low degree of ossification and the dislocation of the elements within the braincase. A dorsoventrally flat, posterodorsal flange of the prootic likely formed the anteromedial portion of the paroccipital process. A foramen on the lateral surface likely reflects the exit of the facial nerve (CN.VII).

The opisthotic is a complex, anteriorly concave element that forms the posterior wall of the otic capsule (Fig 3). The element is sub-triangular in occipital view with a long paroccipital process extending dorsolaterally towards the supratemporal and a broad facet on the ventromedial surface contacting the dorsal process of the exoccipitals. An oval concavity on the floor of the opisthotic likely reflects either the sacculus or the lagenar recess of the inner ear. Due to the taphonomic distortion of the specimen, the margins of the fenestra vestibuli cannot be discerned. An embayment on the anteroventral margin of the opisthotic may represent part of a pressure relief window as seen in *Macroleter* [10]; however, this identification should be considered tentative due to the poorer degree of preservation of this region in *Emeroleter* compared to *Macroleter*.

The supraoccipital is a relatively small element, only extending to the lateral edge of the exoccipital (Fig 3). It is roughly rectangular in dorsal view with no distinct processes and faint indications of a low medial sagittal ridge, as seen in many other parareptiles. The ventral surface appears to bear paired concavities which may represent endolymphatic fossae [25,26].

**Mandible.** The mandibular ramus of *Emeroleter* is an elongate structure that can be subdivided into three main regions: an anterior tooth bearing region, a tall coronoid region, and a slender posterior region including the articular condyles.

The dentary is the only tooth-bearing element of the mandibular ramus and comprises most of its lateral margin, extending from the symphysis anteriorly to the level of the posterior end of the coronoid eminence (Fig 4). The majority of the ventral border of the dentary shares a long, straight suture with the splenial; the posteriormost portion is excluded from the ventral margin of the mandible by the angular and surangular, while the posterodorsal margin contacts the coronoid (Fig 4B). The more complete right dentary bears 19 alveoli on its tooth bearing surface, most of which are occupied by simple, slightly recurved monocuspid teeth that terminate at a sharp point. All the teeth are homodont in form, with no area of enlarged dentition present.

The splenial covers most of the medial surface of the dentary, posteriorly contacting the angular and prearticular and extending anteriorly along the entire preserved length of the dentary (Fig 4D). The mandibular symphysis is broken in KPM uncat/E2, so the presence or degree of contribution of the splenial to the symphysis cannot be determined.

The single edentulous coronoid is well defined with a pronounced dorsal process and overlies the posterior end of the dentary (Fig 4). The anterior extent of the coronoid terminates at the level of the posteriormost dentary tooth, and the posterior process of the element contacts the posterior end of the surangular. The tall coronoid eminence is composed entirely of the coronoid itself, with no contribution from the dentary.

The surangular is relatively long, extending anteriorly to near the level of the anterior end of the coronoid eminence in both lateral and medial views (Fig 4). The anterior half of the surangular is overlain by the dentary while the posteriormost portion is overlain by the articular.

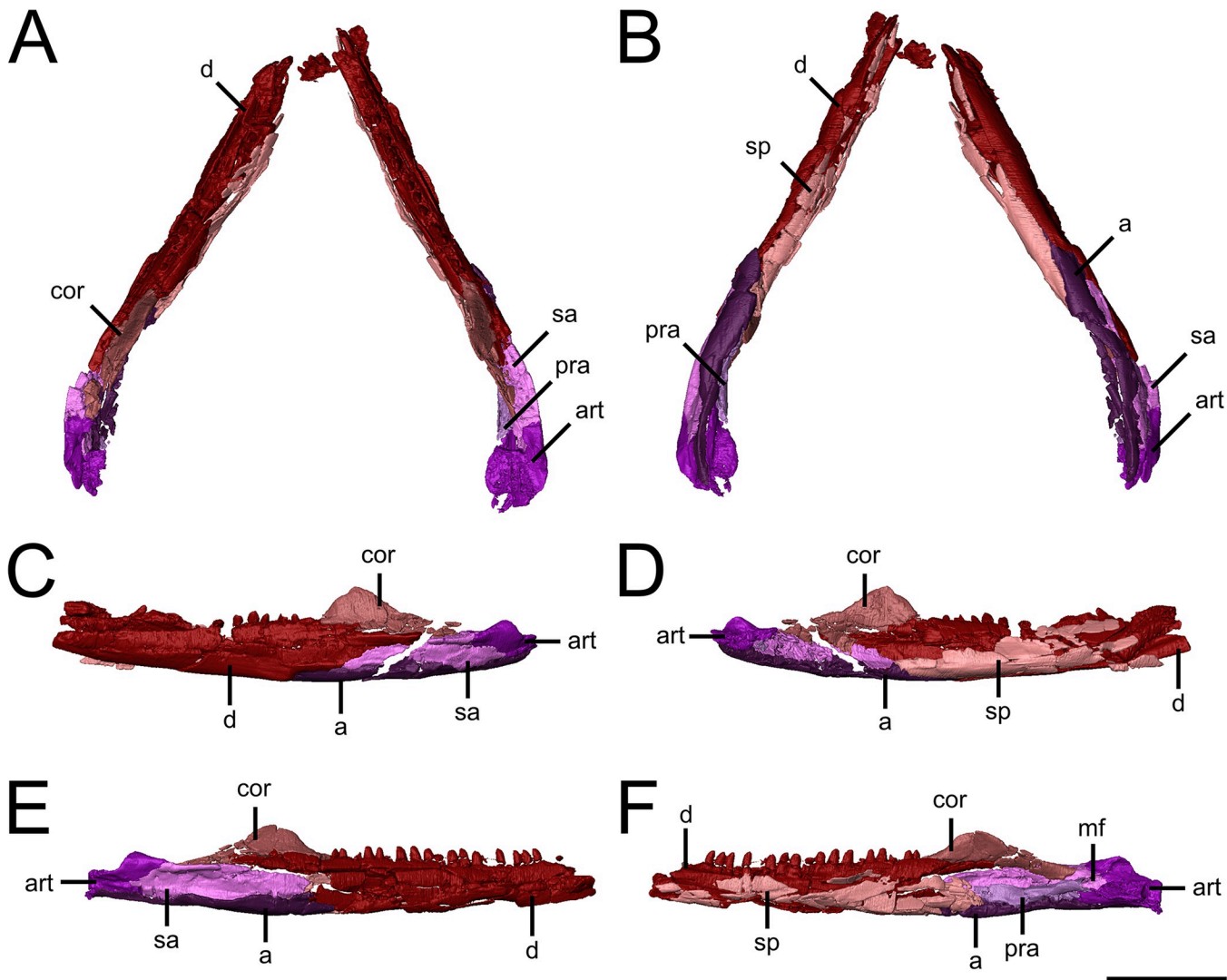

**Fig 4. Lower jaw of the holotype of *Emeroleter levis*. A-B,** lower jaw of KPM uncat/E2 in **A,** dorsal and **B,** ventral views. **C-D,** isolated left mandibular ramus in **C,** labial and **D,** lingual views. **E-F,** isolated right mandibular ramus in **E,** labial and **F,** lingual views. Abbreviations: **a,** angular; **art**, articular; **cor**, coronoid; **d**, dentary; **pra**, prearticular; **sa**, surangular; **sp**, splenial. Scale bar equals 1 cm.

The middle portion of the dorsal margin of the surangular contributes to the dorsal margin of the mandible with a lateral projection of the dorsal margin forming a shelf as in *Macroleter* [9]; this shelf continues posteriorly onto the articular. The surangular also forms the dorsal margin of the Meckelian fossa.

The angular forms the ventral margin of the posterior half of the mandibular ramus, narrowing markedly in its posteriormost portion (Fig 4B). The lateral exposure of the angular is slightly greater than the medial exposure, with a small anterior extension separating the posteriormost portions of the dentary and splenial. The angular is overlain for most of its length by the surangular; the anteriormost section contacts the dentary while the posteriormost section contacts the articular. Ventrally the angular wraps around the mandibular ramus where it contacts the ventral edge of the prearticular and the posterior end of the splenial.

The prearticular is anteroposteriorly long and mediolaterally narrow, and floors the adductor foramen (Fig 4F). The prearticular contacts the surangular laterally, the splenial anteriorly,

the articular posteriorly, and the angular ventrally, and forms the medial margin of the Meckelian fossa.

The articular is a mediolaterally broad, dorsoventrally short element which contacts the surangular and angular along its anteroventral and posteroventral margins, respectively (Fig 4). The element frames a large portion of the Meckelian fossa, forming the posterodorsal, lateral, and ventral margins (Fig 4F). Dorsally it exhibits a continuation of the lateral shelf that is found on the surangular. The articular condyles are dorsally concave and medially slanted; behind the condyle is a short retroarticular process.

## Phylogenetic analysis

The results of the phylogenetic analysis produced 53 most parsimonious trees, each with a tree length of 788. Both the 50% majority rule consensus and strict consensus of these trees recovers *Emeroleter* as the sister taxon to *Nycteroleter* (Fig 5), with this clade being the sister to another that has *'Bashkyroleter' bashkyricus* as the sister taxon to the clade consisting of *'Bashkyroleter' mesensis* and *Rhipaeosaurus*. *Macroleter* is recovered as the sister taxon to the clade which contains all the aforementioned taxa. This completely resolved Nycteroleteridae contrasts with recent studies that find most nycteroleterids in a large polytomy [3–5]. The remaining topology of the 50% majority rule consensus tree (Fig 5A) is identical to what has been recovered by Cisneros et al. [5], which itself is largely similar to previous studies with the exception of the position of *Lanthanosuchus* [3,4], which was found as the sister taxon to the nycteroleters. However, the strict consensus tree recovers some of the other parareptilian clades in a polytomy with the Nycteroleteridae + *Lanthanosuchus* + Pareiasauridae clade (Fig 5B).

## Comparison and discussion

### Palate

The triangular vomers of *Emeroleter* contrast markedly with the rectangular elements of the procolophonid *Procolophon* [27]. While the vomer of *Emeroleter* does bear teeth, it appears to lack the shagreen of denticles seen in other nycteroleters such as *Macroleter* [9], instead having more organized rows. The morphology of the orbitonasal ridge and paraseptal cartilage are very similar to that seen in *Macroleter* and in captorhinids [9,23]; the contribution of the palatal elements to the orbitonasal ridge does seem to vary, however, with the ridge being contained entirely on the vomer in *Emeroleter* while the palatine forms the posterior portion of the ridge in *Macroleter* [9]. The condition in *Emeroleter* in which the orbitonasal ridge, and the depression for the paraseptal cartilage, is housed entirely on the vomer, is also seen in the procolophonid *Procolophon* [27]. The condition in *Emeroleter* where the vomers are in contact medially for most of their length is also seen in the procolophonoid *Saurodektes* [28,29]; in *Delorhynchus*, the vomers are wedged apart in their posterior portion by an anterior process of the pterygoid [24]. The paired tooth-bearing ridges of the vomer, one along the midline and one along the lateral edge parallel to the maxilla, are also shared with *Saurodektes* [28].

The morphology of the ridges in *Emeroleter* is almost identical to *Saurodektes*, with the median ridge continuing posteriorly along the midline of the pterygoid and the lateral ridge continuing along the lateral edge of the palatine before turning posteromedially at approximately the anteroposterior midpoint of the palatine [28]. The lateral ridge eventually converges with the median ridge near the edge of the interpterygoid vacuity, just anterior to the base of the transverse process of the pterygoid [28]. In the pareiasaur *Embrithosaurus*, the median ridge is paired with a second ridge positioned immediately lateral to it [30].

The contribution of the palatine to the posterior border of the choana in *Emeroleter* is common among parareptiles, also being seen in *Macroleter*, the procolophonid *Procolophon*, and

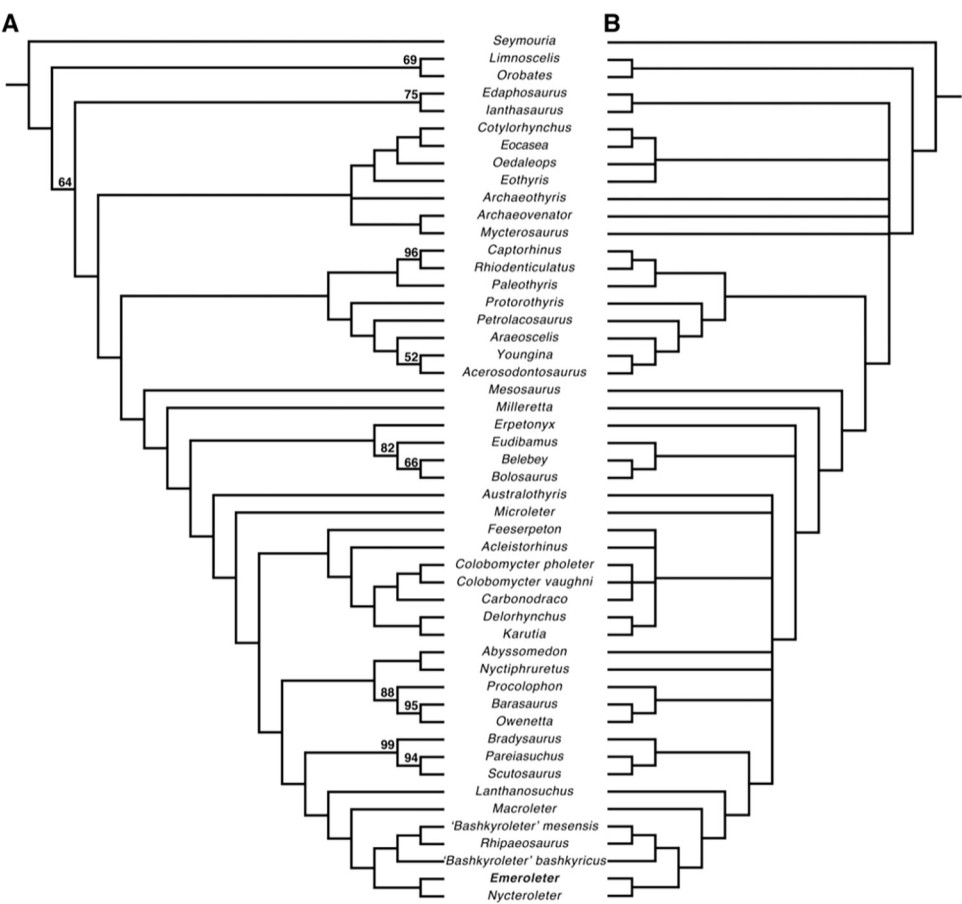

**Fig 5. Consensus trees produced from the 53 optimal trees recovered from the phylogenetic analysis. A**, 50% majority rule consensus tree and **B**, strict consensus tree. Tree length = 788, Consistency index (CI) = 0.291, Rescaled CI = 0.182, Retention index = 0.625. Bootstrap support values above 50% are found above nodes.

the pareiasaurs *Embrithosaurus* and *Deltavjatia* [9,27,31]. The palatine is less heavily denticulated than in *Delorhynchus* [24]. In *Emeroleter*, the palatine ascends anterolaterally to contact the lacrimal; a similar, though not identical, contact is seen in the bolosaurid *Belebey* [11].

The presence of dentition on the ectopterygoid of *Emeroleter* is unusual as the element is edentulous in most parareptiles [9,30,31].

The morphology of the pterygoid of *Emeroleter* is similar to that of *Macroleter* in many aspects, including the strongly recurved transverse flange oriented almost directly laterally [9]; in pareiasaurs the transverse flange is much less recurved, being more anterolaterally directed [31]. While the transverse flange bears the large teeth in *Emeroleter*, *Macroleter*, and the acleistorhinid *Colobomycter*, in the procolophonoids *Saurodeketes* and *Procolophon* and the bolosaurids *Belebey* and *Eudibamus* the flange is entirely edentulous [4,9,27,28,32]. The denticulate median ridges on the palatal ramus running alongside the interpterygoid vacuity and continuing anteriorly onto the vomer are shared with *Macroleter* and *Saurodektes* [9,28]. The short, squared interpterygoid vacuity differs from the narrow condition in *Delorhynchus* [24], the highly elongated vacuity in the acleistorhinid *Feeserpeton* [33], and the more oval condition in *Procolophon* [27]. The contribution of the pterygoid and parabasisphenoid to the margins of the vacuity in *Emeroleter* are like that described in the pareiasaur *Embrithosaurus* [30].

## Braincase

The absence of a cultriform process on the parabasisphenoid of *Emeroleter* is a trait shared among all nycteroleters [9,13], differing from the narrow wedge of *Delorhynchus* [24], the elongated process of *Feeserpeton* [33], and the broad flat process of *Embrithosaurus* [30]. Highly reduced processes are known in the pareiasaur *Deltavjatia* and the procolophonid *Procolophon* [31,34]. The anterolateral direction of the basipterygoid processes differs from *Macroleter* in which they project directly anteriorly [9]; however, the immobility of the basicranial articulation is shared with *Macroleter* and the pareiasaur *Deltavjatia* [9,31]. The blade-like clinoid processes of *Emeroleter* differ substantially from the rounded condition in *Macroleter* [9]. While the hourglass shape to the element in ventral view is shared with *Macroleter* and the pareiasaurs *Embrithosaurus* and *Deltavjatia* [9,30,31], it differs from the wedge-shaped element seen in *Delorhynchus*, *Belebey*, *Eudibamus*, *Colobomycter*, and *Feeserpeton* [4,11,24,35].

The inclusion of the basioccipital in the foramen magnum differs from *Macroleter*, *Saurodektes*, *Leptopleuron*, and *Belebey* in which the element is excluded by the ventral processes of the exoccipitals [9,11,28,36]. Whereas the basioccipital forms the entirety of the occipital condyle in *Emeroleter*, it only forms the ventral portion of the condyle in *Macroleter*, the procolophonid *Leptopleuron*, and in pareiasaurs [9,31,36]. The dorsolaterally elevated lateral edges which contribute to the rim of the fenestra vestibuli are also seen in *Leptopleuron* [36].

The exoccipitals of *Emeroleter* differ markedly from those of *Macroleter*, lacking the lateral projection of the dorsal process seen in the latter taxon [9]; in *Macroleter* and in pareiasaurs, the lateral projection extends for a substantial distance under the paroccipital process of the opisthotic [9,31].

The prootic is a relatively small element in *Emeroleter*, compared with the much more massive structure in *Feeserpeton* and *Embrithosaurus* [30,33]. The contact between the anteroventral portion of the prootic and the clinoid processes of the parabasisphenoid shared with *Macroleter* and most procolophonids [9,36], although *Leptopleuron* lacks this contact [36]. The contribution of the prootic to the anterior portion of the paroccipital process is also seen in *Macroleter*, the procolophonid *Feeserpeton*, and the pareiasaur *Deltavjatia* [9,33].

The opisthotics in *Emeroleter* are relatively small compared with *Feeserpeton* in which they comprise a much greater proportion of the occiput [33]. In *Macroleter* and the pareiasaurs *Embrithosaurus* and *Deltavjatia*, the paroccipital process extends to contact the squamosal and the supratemporal [9,30]; the process appears to have reached the supratemporal in *Emeroleter* but the low degree of ossification precludes identification of a contact with the squamosal. The dorsolateral direction of the paroccipital process is shared with *Macroleter* [9], in contrast with the more directly lateral orientation as in the owenettid *Saurodektes* [28].

The median sagittal ridge of the supraoccipital is common among parareptiles [9] but is much less pronounced in *Emeroleter* than in *Feeserpeton* [35]. The supraoccipital of *Emeroleter* is unusual in being excluded from the foramen magnum; in *Macroleter*, the procolophonid *Leptopleuron*, and the pareiasaur *Embrithosaurus*, the element forms the dorsal margin of the foramen [9,30,36]. The flat, plate-like supraoccipital of *Emeroleter* contrasts markedly with the tall, columnar supraoccipital of pareiasaurs [31,37,38].

## Mandible

While the dentary forms the anterior half of the ventral surface of the mandible in *Emeroleter*, it makes no contribution to the ventral surface at all in *Embrithosaurus* [30]. We cannot determine with certainty whether the splenial was involved in the symphysis in *Emeroleter* due to the loss of the anteriormost portion of the mandible, but splenial contributions to the symphysis are known in *Feeserpeton* and the pareiasaurs *Embrithosaurus* and *Deltavjatia* [30,31,33].

The splenial is excluded from the mandibular symphysis in *Macroleter*, the procolophonoids *Saurodektes* and *Procolophon*, the acleistorhinid *Colobomycter*, and the bolosaurid *Belebey* [4,9,11,27,28]. We were unable to identify the foramen intermandibularis caudalis in *Emeroleter* due to damage to the specimen, but the foramen is typically positioned at the intersection of the splenial, prearticular, and angular in most early reptiles [33,39]; however, some variation is present, as the foramen is situated between the splenial anteriorly and the prearticular posteriorly in the procolophonid *Sauropareion* [40].

The presence of a single coronoid is shared with *Macroleter*, the acleistorhinid *Feeserpeton*, and the pareiasaur *Deltavjatia* [9,33], whereas multiple coronoids have been documented in *Delorhynchus* [39]. The edentulous coronoid in *Emeroleter* differs from the denticulate condition seen in *Feeserpeton* and *Delorhynchus* [33,39]. While in *Emeroleter* the coronoid process is formed solely by the coronoid bone itself, in *Delorhynchus* and *Acleistorhinus* the process includes a contribution from the surangular [24,41]. The anteroposteriorly short, dorsoventrally tall coronoid in *Emeroleter* differs from the lower, more elongated element in *Feeserpeton* [33]; less developed processes are also seen in *Embrithosaurus* [30]. Although the process in *Emeroleter* is well developed, it is more rounded than the tall, pointed process of the bolosaurid *Belebey* [11].

## Phylogenetic implications

Past phylogenetic analyses of parareptiles have often recovered a largely unresolved Nycteroleteridae, with *Macroleter* most often found as the sister to an unresolved polytomy of all other nycteroleters [3–5]. This is likely in part due to there being few modern studies concentrating on and describing nycteroleterid fossil material [9,10,13,15], and none that make use of computed tomography data.

The phylogenetic results of this study produce a fully resolved Nycteroleteridae (Fig 5), granting us much needed resolution to the clade. One of the most immediately obvious implications of this resolved topology is that *'Bashkyroleter' mesensis* and *'Bashkyroleter' bashkyricus* are not sister taxa. This further supports the results of the analyses conducted by Tsuji et al. [13], which identified the paraphyletic nature of the genus *'Bashkyroleter'*, though they were only able to recover a resolved Nycteroleteridae using Bayesian analysis. Due to poor support for this paraphyletic relationship they did not erect a new genus for *'Bashkyroleter' mesensis*. For similar reasons we also do not erect a new genus for this taxon. Furthermore, while we also recover these taxa as paraphyletic, we find different positions for them than what was recovered in the analyses of Tsuji et al. [13], with '*Bashkyroleter' mesensis* being found as the sister to *Rhipaeosaurus* and *'Bashkyroleter' bashkyricus* is recovered as the sister to that clade (Fig 5).

Lastly, we recover the enigmatic *Lanthanosuchus watsoni* as the sister taxon of Nycteroleteridae, confirming the topology found by Cisneros et al. [5]. *Lanthanosuchus* has long been a problematic taxon due to its distinctive morphology in comparison to other parareptiles, but for the past two decades has been considered a parareptile [41]. Further work will have to be done to better understand its position among early tetrapods, in particular examination of the taxon using computed tomography will likely be important for accomplishing this.

## Functional implications

Prior to the description of an impedance-matching ear in *Macroleter* [10], impedance-matching was unknown in any Palaeozoic amniote. Müller and Tsuji [10] identified several features in *Macroleter* that suggested the presence of an impedance-matching ear, including (i) a tympanum-to-stapedial-footplate ratio well within the range of modern amniotes with

sophisticated auditory apparatuses, (ii) a small fenestra ovalis (reduced from an ancestrally large fenestra vestibuli), (iii) a possible pressure relief window, (iv) a well-ossified medial wall of the otic capsule, and (v) a short, slender stapes directed towards, but not reaching, the hypothesized tympanic membrane. Due to the immaturity of KPM uncat/E2 and the dorso-ventral crushing of the specimen, many of these features cannot be evaluated in *Emeroleter*, including the tympanum-to-footplate area ratio, the ossification of the medial wall, or the morphology of the stapes or fenestra ovalis/fenestra vestibuli. However, two notable features are identifiable in *Emeroleter*. First, the morphology of the otic notch in *Emeroleter* is strikingly similar to that of *Macroleter*, both in terms of relative size and the elements contributing to the notch, suggesting a similarly large tympanum [13]. Given that all known nycteroleterid stapes correspond to the morphology seen in *Macroleter* [10], there is no reason to expect *Emeroleter* to differ in this regard. If *Emeroleter* did possess a similarly short and slender stapes as that seen in other nycteroleterids, this morphology combined with the extremely large tympanum would suggest a hearing acuity at least equivalent to that estimated in *Macroleter*, which was itself suggested to be roughly comparable to that of the average modern lizard [10]. Second, the tentatively identified pressure relief window in *Emeroleter* is similar in both placement and general morphology to that of *Macroleter*, being positioned just anterior to the metotic foramen and bounded by somewhat irregular margins (see Fig 4 in [10]). Overall, the morphology seen in *Emeroleter* is largely similar to that in *Macroleter*. The probable presence of impedance-matching hearing in multiple taxa that are not closely related presents two possible evolutionary scenarios: an impedance-matching hearing system may be more widely present throughout nycteroleterids, or impedance-matching may have arisen multiple times within the clade. Given that *Macroleter* is the basalmost nycteroleterid while *Emeroleter* is recovered as highly nested within Nycteroleteridae, and given the probable presence of a tympanum in all nycteroleterids (with the exception of *Rhiphaeosaurus* which is too incomplete to allow for identification of the presence or absence of a tympanum), we consider the former scenario to be the most likely. However, as multiple independent origins of impedance-matching hearing are known throughout crown tetrapods [42], more neurocranial information on a wider range of nycteroleterids, and parareptiles more broadly, will be necessary to more confidently determine which scenario is most probable.

## Supporting information

**S1 File. Data matrix for phylogenetic analyses of Nycteroleteridae.**
(NEX)

## Acknowledgments

Thanks to the Vyatka Paleontological Museum for access to the specimen. We also thank Juan Cisneros and an anonymous reviewer for their constructive comments.

## Author Contributions

**Conceptualization:** Jörg Fröbisch.

**Formal analysis:** Kayla D. Bazzana-Adams, Mark J. MacDougall.

**Funding acquisition:** Jörg Fröbisch.

**Investigation:** Kayla D. Bazzana-Adams, Mark J. MacDougall.

**Methodology:** Kayla D. Bazzana-Adams, Mark J. MacDougall.

**Project administration:** Jörg Fröbisch.

**Resources:** Jörg Fröbisch.

**Software:** Jörg Fröbisch.

**Supervision:** Jörg Fröbisch.

**Visualization:** Kayla D. Bazzana-Adams, Mark J. MacDougall.

**Writing – original draft:** Kayla D. Bazzana-Adams, Mark J. MacDougall.

**Writing – review & editing:** Kayla D. Bazzana-Adams, Mark J. MacDougall, Jörg Fröbisch.

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
