## [Decision Letter · Decision Letter 0]

4 Dec 2023

PONE-D-23-33852Cranial anatomy of Emeroleter levis and the phylogeny of NycteroleteridaePLOS ONE

Dear Dr. Bazzana-Adams,

Thank you for submitting your manuscript to PLOS ONE. After careful consideration, we feel that it has merit but does not fully meet PLOS ONE’s publication criteria as it currently stands. Therefore, we invite you to submit a revised version of the manuscript that addresses the points raised during the review process.

Please read carefully on the comments and revise your paper.

We look forward to receiving your revised manuscript.

Kind regards,

Jun Liu

Academic Editor

PLOS ONE

Journal Requirements:

2. In your manuscript, please provide additional information regarding the specimens used in your study. Ensure that you have reported human remain specimen numbers and complete repository information, including museum name and geographic location. 

For more information on PLOS ONE's requirements for paleontology and archeology research, see https://journals.plos.org/plosone/s/submission-guidelines#loc-paleontology-and-archaeology-research.

   "This work was supported by funding from the German Research Foundation (DFG project No. 2457/9-1) and a scholarship to KBA from the Humboldt Internship Program."

Additional Editor Comments:

One problem is related to the figures:many discussed features not marked in figures

Figure 1 take out of mandibles from ventral view

L170 in my view, only posterior margin

L190 groove, mark on figure 2

cp also on figure 2,

L240 oval concavity mark it on Fig 3

L274 The surangular is relatively long, extending anteriorly to the level of the anterior end of the coronoid eminence. [No, it is posterior to that]

L280 MARK foramen intermandibularis caudalis on fig4, I cannot see it

L295 MATK foramen intermandibularis on Fig 4A

orbitonasal ridge and paraseptal cartilage mark on fig2

meckelian fossa is unclear in your figure 4, mark it

Reviewers' comments:

Reviewer's Responses to Questions

**Comments to the Author**

1. Is the manuscript technically sound, and do the data support the conclusions?

Reviewer #1: Partly

Reviewer #2: Yes

2. Has the statistical analysis been performed appropriately and rigorously? 

Reviewer #1: I Don't Know

Reviewer #2: Yes

3. Have the authors made all data underlying the findings in their manuscript fully available?

Reviewer #1: Yes

Reviewer #2: Yes

4. Is the manuscript presented in an intelligible fashion and written in standard English?

Reviewer #1: Yes

Reviewer #2: Yes

5. Review Comments to the Author

Reviewer #1: The paper “Cranial anatomy of Emeroleter levis and the phylogeny of the Nycteroleteridae” is an interesting paper with new and important morphological data from a poorly-known yet not-insignificant clade. The methodology for the CT scanning and segmentation seems to be sound and the resulting figures and description are valuable.

Only a couple of typos of note: Line 110: “boarder” should be “border”; Line 446-7: “examination of the taxon” should be “examination of additional taxa” or the like.

The anatomical description is complete; it may be a personal preference for a single comparative description within the results, rather than having the comparisons contained separately within the discussion. However, if the comparison is to feature in the discussion, I think it would add to the relevance of the paper if this was done within a phylogenetic framework. Is any of this morphology relevant to the phylogenetic analysis completed in the paper?

However; there are major issues in the phylogenetic analysis and systematic palaeontology that should be addressed before acceptance and publication.

Phylogenetic analysis: Usually at least 10 000 replicates are done in the heuristic search. More replicates would not greatly increase the analysis time, but would result in a better analysis. Also a Bayesian analysis would not be out of place here if you are comparing results to papers that use the method.

Also, some of the characters that have been re-scored for Emeroleter are based off this single specimen, which is likely juvenile and also likely crushed dorso-ventrally: ie. rescoring “snout shape wider than tall” when the character scoring was based on other specimens may not be valid. Also, some traits seem to be variable, such as “jugal anterior process extends to al least the level of the anterior orbital rim”, so should be scored as variable.

Revised diagnosis: the characters that are named as autapomorphies of Emeroleter are not, in fact, autapomorphies of the taxon. For example, “maxilla extends to and contributes to the posterodorsal margin of the naris” is common to all ‘nycteroleters’. Please re-evaluate all characters in this section.

Referred specimen: mention that the specimen is likely juvenile and has experienced dorso-ventral crushing.

Phylogenetic Analysis: Please show the results of the strict consensus in the analysis. There are a number of papers (ie. Sharkey and Leathers 2001 in Cladistics, others) that show majority rules consensus is “inappropriate” – just because one topology occurs more often doesn’t mean it is “more correct”. Plus you cannot compare the 50% majority-rules consensus tree to the strict consensus trees of other papers. This analysis had 53 MPTs, so a strict consensus tree would have unresolved branches seen in other analyses. This also means that the “Phylogenetic implications” section needs to be re-written with this in mind. There are only fully-resolved clades because of the use of a consensus tree where other papers did not display the data this way.

General comments: Is there anything in the new data from the braincase anatomy that supports or refutes the hypothesis that nycteroleters had particularly good hearing (Mueller and Tsuji 2007)? I also think the final sentence need to be re-written/be stronger.

Figures: Figures are good, but it is difficult to perceive depth particularly in the dorsal and ventral views (Figure 1 A and B); is there a setting on Amira that can ‘add depth’ to the image? Figure 5: the ‘Bashkyroleters’ should be in single quotes as even here they aren’t monophyletic.

Reviewer #2: This work provides detailed anatomical information on a Permian reptile from Russia. One of the main results is an improved phylogenetic placement based on the most recent version of the data matrix. The work is well written, the results are solid and the figures are of great quality. I have no other comments to make, except two minor ones:

I replicated the phylogenetic analysis on TNT with the data base and the parameters provided in the text. The program found 53 trees as well, but reported a shorter length (783) which differs from the length reported here (787). The topology of the consensus tree, however, is identical. Please double check the tree length.

The taxon "Owenetta kitchingorum" mentioned in the analysis is now considered a junior synonym of Saurodektes kitchingorum (Hamley et al. 2020: https://doi.org/10.1093/zoolinnean/zlaa056). Owenetta is also mentioned several times in comparisons within the text. The authors should check if they are talking about the Permian Owenetta rubidgei or the Triassic Saurodektes kitchingorum (=Owenetta kitchingorum).

Great work!

6. PLOS authors have the option to publish the peer review history of their article (what does this mean?). If published, this will include your full peer review and any attached files.

Reviewer #1: No

Reviewer #2: **Yes: **Juan Carlos Cisneros

---

## [Author Response · Author response to Decision Letter 0]

18 Jan 2024

Full response to reviewer and editor comments is contained in the Response to Reviewers file.

---

## [Editor Report · Decision Letter 1]

22 Jan 2024

Cranial anatomy of Emeroleter levis and the phylogeny of Nycteroleteridae

PONE-D-23-33852R1

Dear Dr. Bazzana-Adams,

We’re pleased to inform you that your manuscript has been judged scientifically suitable for publication and will be formally accepted for publication once it meets all outstanding technical requirements.

Kind regards,

Jun Liu

Academic Editor

PLOS ONE